# Viral genome structures are optimal for capsid assembly

**Jason D Perlmutter, Cong Qiao, Michael F Hagan***

Martin A Fisher School of Physics, Brandeis University, Waltham, United States

**Abstract** Understanding how virus capsids assemble around their nucleic acid (NA) genomes could promote efforts to block viral propagation or to reengineer capsids for gene therapy applications. We develop a coarse-grained model of capsid proteins and NAs with which we investigate assembly dynamics and thermodynamics. In contrast to recent theoretical models, we find that capsids spontaneously 'overcharge'; that is, the negative charge of the NA exceeds the positive charge on capsid. When applied to specific viruses, the optimal NA lengths closely correspond to the natural genome lengths. Calculations based on linear polyelectrolytes rather than base-paired NAs underpredict the optimal length, demonstrating the importance of NA structure to capsid assembly. These results suggest that electrostatics, excluded volume, and NA tertiary structure are sufficient to predict assembly thermodynamics and that the ability of viruses to selectively encapsidate their genomic NAs can be explained, at least in part, on a thermodynamic basis.

## Introduction

For many viruses the spontaneous assembly of a protein shell, or capsid, around the viral nucleic acid (NA) is an essential step in the viral lifecycle. Identifying the factors which enable capsids to efficiently and selectively assemble around the viral genome could identify targets for new antiviral drugs that block or derail the formation of infectious virions. Conversely, understanding how assembly depends on the NA and protein structure would guide efforts to reengineer capsid proteins and human NAs for gene therapy applications. From a fundamental perspective, high-order complexes that assemble from protein and/or NAs abound in biology. Learning how the properties of viral components determine their co-assembly can shed light on assembly mechanisms of a broad array of structures and the associated selective pressures on their components. In this article, we use GPU computing (*Anderson et al., 2008*; *Nguyen et al., 2011*; *LeBard et al., 2012*) and a simplified, but quantitatively testable, model to elucidate the effects of electrostatics, capsid geometry, and NA tertiary structure on assembly.

Assembly around NAs is predominately driven by electrostatic interactions between NA phosphate groups and basic amino acids, often located in flexible tails known as arginine rich motifs (ARMs) (e.g., *Schneemann, 2006*). There is a correlation between the net charge of these protein motifs and the genome length for many ssRNA viruses (*Belyi and Muthukumar, 2006*; *Hu et al., 2008*), with a 'charge ratio' of negative charge on NAs to positive charge on proteins typically of order 2:1 (i.e., viruses are 'overcharged'). Electrophoresis measurements confirm that viral particles are negatively charged (e.g., [*Serwer et al., 1995*; *Serwer and Griess, 1999*; *Porterfield et al., 2010*]), though these measurements include contributions from the capsid exteriors (*Bozic et al., 2012*; *Zlotnick et al., 2013*). Based on these observations, it has been proposed that viral genome lengths are thermodynamically optimal for assembly, meaning that their lengths minimize the free energy of the assembled nucleocapsids. However, while estimates of optimal lengths have varied (*van der Schoot and Bruinsma, 2005*; *Angelescu et al., 2006*; *Belyi and Muthukumar, 2006*; *Hu et al., 2008*; *Siber and Podgornik, 2008*; *Ting et al., 2011*; *Ni et al., 2012*; *Siber et al., 2012*), recent theoretical models based on linear polyelectrolytes (*Siber and Podgornik, 2008*; *Ting et al., 2011*; *Ni et al., 2012*) have consistently predicted that optimal NA lengths correspond to 'undercharging'

*For correspondence: hagan@brandeis.edu

Competing interests: The authors declare that no competing interests exist.

**eLife digest** Viruses are infectious agents made up of proteins and a genome made of DNA or RNA. Upon infecting a host cell, viruses hijack the cell's gene expression machinery and force it to produce copies of the viral genome and proteins, which then assemble into new viruses that can eventually infect other host cells. Because assembly is an essential step in the viral life cycle, understanding how this process occurs could significantly advance the fight against viral diseases.

In many viral families, a protein shell called a capsid forms around the viral genome during the assembly process. However, capsids can also assemble around nucleic acids in solution, indicating that a host cell is not required for their formation. Since capsid proteins are positively charged, and nucleic acids are negatively charged, electrostatic interactions between the two are thought to have an important role in capsid assembly. However, it is unclear how structural features of the viral genome affect assembly, and why the negative charge on viral genomes is actually far greater than the positive charge on capsids. These questions are difficult to address experimentally because most of the intermediates that form during virus assembly are too short-lived to be imaged.

Here, Perlmutter et al. have used state of the art computational methods and advances in graphical processing units (GPUs) to produce the most realistic model of capsid assembly to date. They showed that the stability of the complex formed between the nucleic acid and the capsid depends on the length of the viral genome. Yield was highest for genomes within a certain range of lengths, and capsids that assembled around longer or shorter genomes tended to be malformed.

Perlmutter et al. also explored how structural features of the virus—including base-pairing between viral nucleic acids, and the size and charge of the capsid—determine the optimal length of the viral genome. When they included structural data from real viruses in their simulations and predicted the optimal lengths for the viral genome, the results were very similar to those seen in existing viruses. This indicates that the structure of the viral genome has been optimized to promote packaging into capsids. Understanding this relationship between structure and packaging will make it easier to develop antiviral agents that thwart or misdirect virus assembly, and could aid the redesign of viruses for use in gene therapy and drug delivery.

(fewer NA charges than positive capsid charges). These results lead to the conclusion that capsid assembly around genomic (overcharged) NAs requires an external driving force such as a Donnan potential (*Ting et al., 2011*). Yet, viruses preferentially assemble around genomic length RNAs even in vitro (*Comas-Garcia et al., 2012*), in the absence of such a driving force.

The effect of NA structural features other than charge remains unclear. In some cases, genomic NAs are preferentially packaged over others with equivalent charge (*Borodavka et al., 2012*) due to virus-specific packaging sequences (*Bunka et al., 2011*; *Borodavka et al., 2012*). However, experiments on other viruses have demonstrated a striking lack of virus-specific interactions (*Porterfield et al., 2010*; *Comas-Garcia et al., 2012*). For example, cowpea chlorotic mottle virus (CCMV) proteins preferentially encapsidate BMV RNA over the genomic CCMV RNA (*Comas-Garcia et al., 2012*). Since the two NAs are of similar length, the authors propose that other structural features, such as NA tertiary structure (*Yoffe et al., 2008*), may drive this preferential encapsidation. However, the relationship between NA structure and assembly has not been explored.

To clarify this relationship, we use a computational model to investigate capsid assembly dynamics and thermodynamics as functions of NA and capsid charge, solution ionic strength, capsid geometry, and NA size (resulting from tertiary structure). We first test the proposed link between the thermodynamic optimum length, $L^*_{eq}$ and assembly, finding that the yield of assembled nucleocapsids at relevant timescales is maximal near $L^*_{eq}$. Longer-than-optimal NAs lead to non-functional structures, indicating that the thermodynamic optimum $\left(L^*_{eq}\right)$ corresponds to an upper bound for the genome size for capsids which spontaneously assemble and package their genome. We then explore how $L^*_{eq}$ depends on solution conditions and the structures of capsids and NAs. We find that overcharging occurs spontaneously, requiring no external driving force. When base-pairing is accounted for, predicted optimal NA lengths are consistent with the genome size for a number of viruses, suggesting that electrostatics and NA tertiary structure are important factors in the formation and stability of viral particles.

Our predictions can be tested quantitatively in in vitro packaging experiments (e.g., [*Porterfield et al., 2010*; *Cadena-Nava et al., 2012*; *Comas-Garcia et al., 2012*]).

## Model

Our coarse-grained capsid model (*Figure 1*) is motivated by the recent observation (*Kler et al., 2012*) that purified simian virus 40 (SV40) capsid proteins assemble around ssRNA molecules in vitro to form capsids comprising 12 homopentamer subunits. We model the capsid as a dodecahedron, composed of 12 pentagonal subunits (each of which represents a rapidly forming and stable pentameric intermediate, which then more slowly assembles into the complete capsid, as is the case for SV40 [*Li et al., 2002*]). Our model extends those of *Wales (2005)*, *Fejer et al. (2009)*, *Johnston et al. (2010)*, with subunits attracted to each other via attractive pseudoatoms at the vertices (type 'A') and driven toward a preferred subunit–subunit angle by repulsive 'Top' pseudoatoms (type 'T') and 'Bottom' pseudoatoms

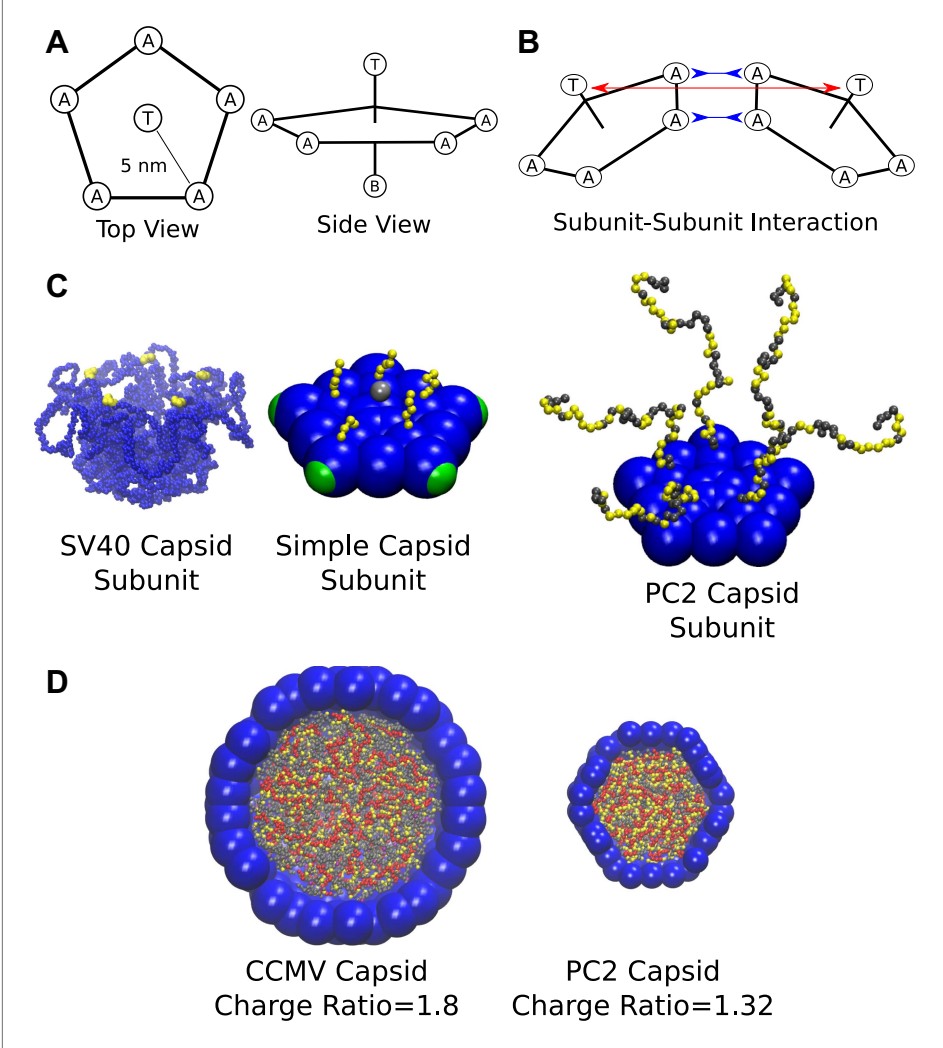

**Figure 1**. Schematics and representative images of model systems. (**A**), (**B**) Model schematic for (**A**) a single subunit, and (**B**) two interacting subunits, showing positions of the attractor ('A'), Top ('T'), and Bottom ('B') pseudoatoms, which are defined in the 'Model' section and in the 'Methods'. (**C**) (left) The pentameric SV40 capsid protein subunit, which motivates our model. The globular portions of proteins are shown in blue and the beginning of the NA binding motifs (ARMs) in yellow, though much of the ARMs are not resolved in the crystal structure (*Stehle et al., 1996*). Space-filling model of the basic subunit model (middle) and a pentamer from the PC2 model (right). (**D**) A cutaway view of complete CCMV and PC2 capsids (with respective biological charge ratios of 1.8 and 1.32). Beads are colored as follows: blue = excluders, green = attractors, yellow = positive ARM bead, gray = neutral ARM bead, red = polyelectrolyte.

(type 'B') (see *Figure 1* and the 'Methods'). In contrast to previous models for polyelectrolyte encapsidation (*Angelescu et al., 2006*; *Elrad and Hagan, 2010*; *Kivenson and Hagan, 2010*; *Mahalik and Muthukumar, 2012*), the proteins contain positive charges located in flexible polymeric tails, representing the ARM (arginine-rich motif) NA binding domains typical of positive-sense ssRNA virus capsid proteins.

To investigate the effect of NA properties on assembly we consider two models for the packaged polymer: (1) a linear flexible polyelectrolyte and (2) a NA with predefined secondary and tertiary structure (i.e., static base-pairs) that captures the size, shape, and rigidity of NAs. Single-stranded regions are modeled as flexible polymers with one bead per nucleotide (*Zhang and Glotzer, 2004*; *ElSawy et al., 2011*), with charge $-e$. Double-stranded regions of NAs comprise two adjoined semiflexible strands with the net persistence length of dsDNA ($\approx 50$ nm), and base-paired nucleotides are connected by harmonic bonds. Electrostatics are modeled using Debye–Huckel interactions to account for screening, except where these are tested against simulations with Coulomb interactions and explicit salt ions (see *Figure 3D* below).

In addition to representing the secondary structures of specific ssRNA genomes, we are able to tune statistical measures of base-pairing, such as the fraction of nucleotides that are base-paired, the relative frequency of hairpins and higher-order junctions (*Figure 6*), and the maximum ladder distance (MLD), which measures the extension in graph space of a NA secondary structure (*Yoffe et al., 2008*). As shown in *Figure 6*, the radius of gyration $R_G$ of the model NAs depends on MLD as $1.7 \times MLD^{0.43}$, which has a slightly smaller exponent than a theory in which only base-paired segments were accounted for (*Yoffe et al., 2008*). Further model details and parameters are presented in the 'Methods'.

## Results

### Capsid assembly leads to spontaneous overcharging

We begin by presenting the results of simulations on our simplest capsid and cargo models. Our model capsid has a dodecahedron inradius (defined as the distance from the capsid center to a face center) of $R_{in} = 7.3$ nm, to give an interior volume consistent with that of the smallest icosahedral viruses, and contains 60 ARMs (i.e., a $T = 1$ capsid, where $T$ is the triangulation number [*Caspar and Klug, 1962*]) each containing five positively charged residues. The cargo is a linear polyelectrolyte. While we systematically alter both the cargo and capsid below to include more biological detail, the simple model demonstrates two important results (that are consistent with results from more complex models): (1) Viral particles spontaneously overcharge during assembly, and (2) The thermodynamic optimal polyelectrolyte length closely correlates with the length for which dynamical assembly leads to the highest yield of complete viral particles.

#### Dynamical simulations

The results of Brownian dynamics simulations of capsid assembly around a linear polyelectrolyte at physiological salt concentration (Debye screening length $\lambda_D = 1$ nm) are shown in *Figure 2*. Consistent with most ssRNA virus proteins, the polymer is essential for assembly under the simulated conditions, since the subunit–subunit interactions are too weak for formation of empty capsids (see below). *Figure 2A* presents representative snapshots of the assembly process for a polyelectrolyte with 600 segments (see also *Video 1*). The subunits first adsorb onto the polymer in a disordered fashion, with on average about eight subunits adsorbing before first formation of a critical nucleus (a complex comprising five subunits, *Figure 2—figure supplement 1*). Once a critical nucleus forms, additional subunits add to it sequentially and reversibly until the final subunit closes around the polymer.

The assembly outcome depends on polymer length, with successful capsid formation occurring when there is overcharging, meaning that the negative charge on the polymer exceeds the net positive charge on an assembled capsid ($300e$ for this model). *Figure 2B* shows the yield of well-formed capsids at $t = 2 \times 10^4 t_u$ ($2 \times 10^8$ time steps), at which point the fraction assembled has approximately plateaued for most parameter values. Here a well-formed capsid is defined as a structure comprising 12 subunits that each interact with five neighboring subunits and together completely encapsulate the polymer. Assembly is robust (yield ~ 0.9) near an optimal polymer length of $L^*_{dyn} = 575$ segments, corresponding to a 'charge ratio' of $575/300 = 1.9$. Above the optimal length, the polymer is typically not fully incorporated when capsid assembly nears completion. For sufficiently long polymers (e.g., $2\,L^*_{dyn}$, *Figure 2B* right)

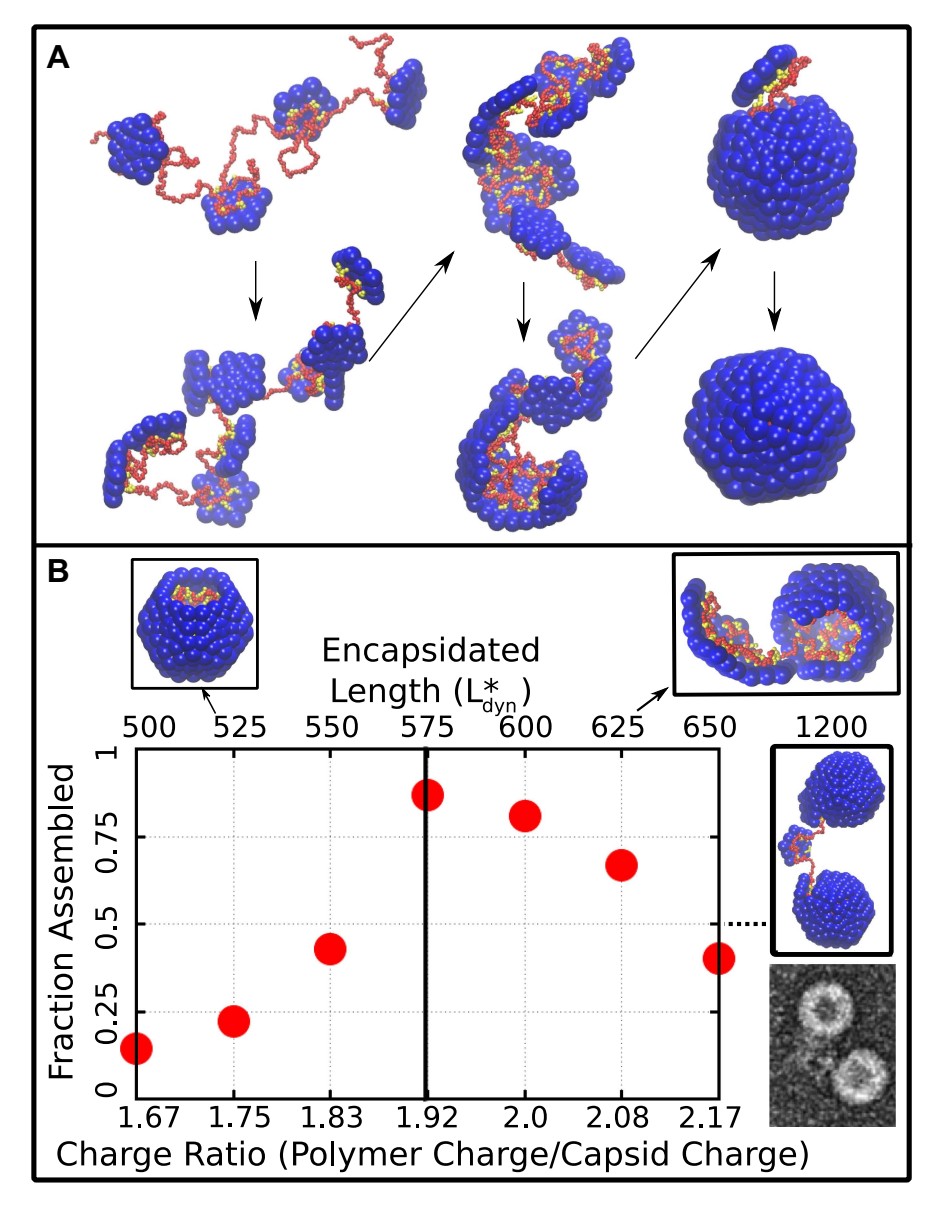

**Figure 2**. Capsid assembly around a linear polyelectrolyte. (**A**) Snapshots illustrating assembly of subunits with ARM length = 5 around a linear polyelectrolyte with 600 segments. Beads are colored as in **Figure 1**. (**B**) Fraction of trajectories leading to a complete capsid as a function of polymer length (top axis) or charge ratio (bottom axis). The dashed line indicates the thermodynamic optimum charge ratio or length ($L^*_{eq}$)from equilibrium calculations. Snapshots of typical outcomes above and below the optimal length are shown. (Far right) A typical assembly outcome for polymer length 1200 (twice $L^*_{eq}$) is compared to an EM image of CCMV proteins assembled around an RNA which is twice the CCMV genome length (image extracted from panel C of Figure 5 in **Cadena-Nava et al., 2012**).

The following figure supplements are available for figure 2:

**Figure supplement 1**. Estimation of the critical nucleus size.

**Figure supplement 2**. The residual chemical potential $\mu_r$ calculated by the Widom test-particle insertion method.

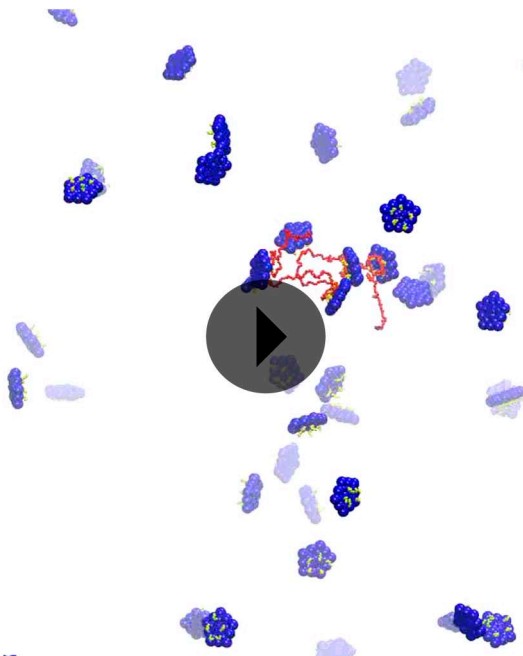

**Video 1**. Capsid Assembly. Movie illustrating assembly of subunits with ARM length=5 around a linear polyelectrolyte with 600 segments. Beads are colored as in *Figure 1*.

multiple capsids assemble on the same polymer. These multiplet structures resemble configurations seen in a previous simulation study which did not explicitly consider electrostatics (*Elrad and Hagan, 2010*) and observed in experiments in which CCMV proteins assembled around RNAs longer than the CCMV genome length (*Cadena-Nava et al., 2012*). For polymer lengths well below $L^*_{dyn}$ the polymer is completely encapsulated before assembly completes, and addition of the remaining subunits slows substantially. Although capsids which are incomplete at the conclusion of these simulation might eventually reach completion, the low yield of assembled capsids at our finite measurement time reflects the fact that assembly at these parameters is less efficient than for polymer lengths near $L^*_{dyn}$.

## Equilibrium calculations

We calculated the thermodynamic optimal polymer length $L^*_{eq}$, or the length of encapsulated polymer that minimizes the free energy of the polymer–capsid complex, with two different methods. First, we performed Brownian dynamics simulations of a long polymer and a preassembled capsid with one subunit made permeable to the polymer, so that the length of encapsulated polymer is free to equilibrate. Second, we calculated the residual chemical potential difference between the encapsidated polymer and a polymer free in solution (*Widom, 1963*; *Kumar et al., 1991*; *Elrad and Hagan, 2010*). The first method predicts an optimal polymer length of $L^*_{eq} = 574$ while the latter suggests $L^*_{eq} \approx 550 - 575$, indistinguishable from the optimal length found in the finite-time dynamical assembly simulations (*Figure 2B*). The observation that the yield of encapsulated polymers from dynamical assembly trajectories diminishes above $L^*_{eq}$, together with the observation that many viruses with single-stranded genomes assemble and package their nucleic acid spontaneously, suggests that this equilibrium value may set an upper bound on the size of a viral genome.

## The effect of control parameters on packaged lengths

### Capsid structure affects packaged lengths

Since our simulations show that $L^*_{eq}$ and $L^*_{dyn}$ are closely correlated, we performed a series of equilibrium calculations in which ionic strength, capsid structural parameters, and the NA model were systematically varied, to determine the effect of each parameter on $L^*_{eq}$. To determine how $L^*_{eq}$ and the optimal charge ratio depend on the number of positive charges in the capsid, we first varied the length of the ARMs, keeping all ARM residues positively charged. As shown in *Figure 3A* (inset), $L^*_{eq}$ increases sub-linearly with capsid charge, meaning that each additional ARM charge increases the equilibrium polymer packaging length by a smaller amount, leading to a diminishing charge ratio. We obtained a similar result when, instead of modeling flexible ARMs, we placed charges in rigid patches on the inner capsid surface (e.g., corresponding to MS2 [*Valegard et al., 1997*]). However, we find that charges on the surface lead to a lower optimal charge ratio than the equivalent number of charges located in flexible ARMs (*Figure 3A*), since the ARM flexibility increases the volume of configuration space available for NA–ARM interactions. These observations demonstrate that, while electrostatics is an important factor, excluded-volume and the lengths of polyelectrolyte segments that bridge between ARMs (discussed below) also affect the length of packaged polyelectrolyte. However, in the biologically relevant range of 5–20 positive charges per protein monomer (*Belyi and Muthukumar, 2006*; *Hu et al., 2008*), the optimal length appears roughly linear with capsid charge (but with a positive intercept).

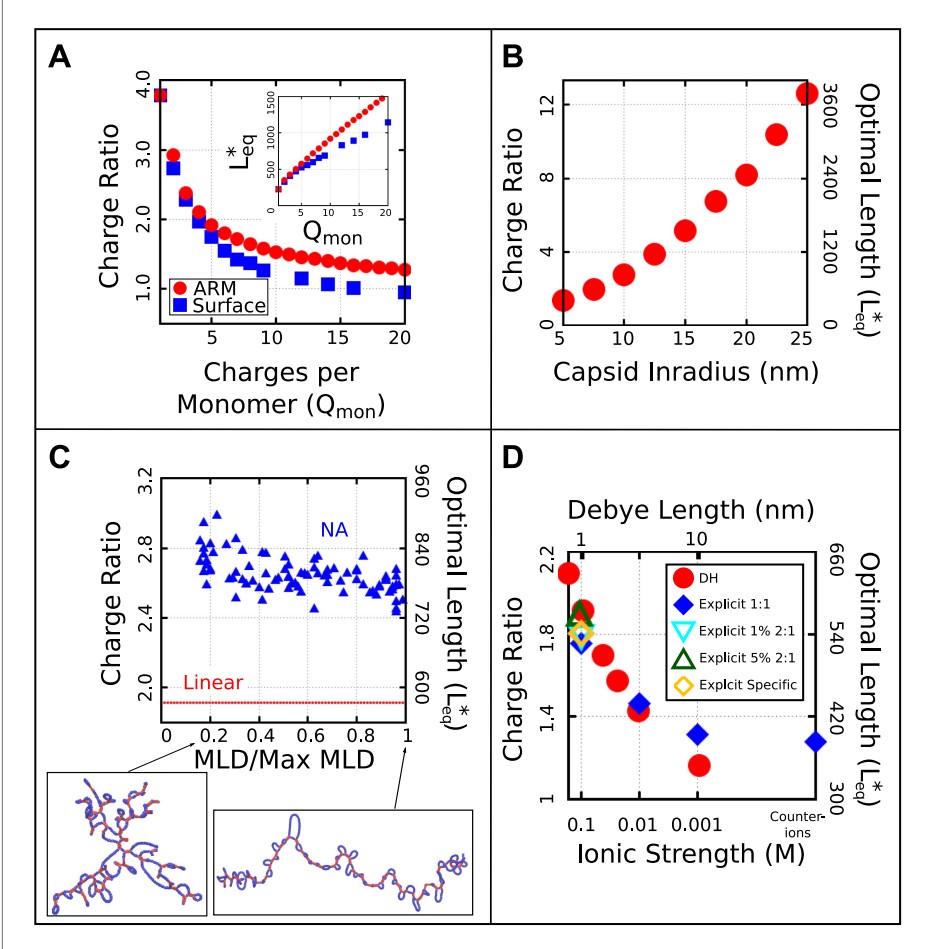

**Figure 3**. Effect of control parameters on the thermodynamic optimal length and charge ratio. (**A**) Effect of increasing capsid charge, with capsid $R_{in}$ = 7.3 nm. (**B**) Effect of increasing capsid size for fixed ARM length = 5. (**C**) Effect of base-pairing, with $f_{bp}$ = 0.5 base-paired nucleotides and varying maximum ladder distance (MLD), for capsid $R_{in}$ = 7.3 nm and ARM Length = 5. Snapshots of our model NA structures with small and large MLD's are shown (prior to encapsidation), with double-stranded regions in red and single-stranded regions in blue. The result for no base-pairing (linear) is shown as a dashed line. (**D**) Effect of ionic strength and comparison between Debye–Huckel interactions and explicit ions. The thermodynamic optimum lengths $L^*_{eq}$ and corresponding optimal charge ratios are shown as functions of the ionic strength (Debye screening length), calculated with simulations using Debye–Huckel (DH) interactions (red circles) or Coulomb interactions with explicit ions, 1:1 salt and no divalent cations (blue diamonds), 1% 2:1 salt (blue triangles), or 5% 2:1 salt (green triangles). An additional system with monovalent free ions and divalent cations irreversibly bound to the polyelectrolyte is also presented (yellow diamonds, see 'Model potentials and parameters'). Calculations were performed using the simple capsid model (**Figure 1**) and a linear polyelectrolyte.

The following figure supplements are available for figure 3:

**Figure supplement 1**. The thermodynamic optimum lengths $L^*_{eq}$ and charge ratios monotonically decrease with increasing persistence length for a linear, semiflexible polyelectrolyte.

**Figure supplement 2**. Effect of varying the ion radius.

**Figure supplement 3**. Effects of counterion condensation on $L^*_{eq}$.

To understand how capsid size influences $L^*_{eq}$, we varied the model capsid radius while holding the number of capsid charges fixed. As shown in **Figure 3B**, $L^*_{eq}$ and hence the optimal charge ratio increase dramatically with capsid size, scaling roughly with capsid radius as $L^*_{eq} \sim R_{in}^{1.6}$. The non-integer exponent is intriguing, as it rules out scaling with capsid volume, surface area, or a linear path length, which

would respectively result in $L^*_{eq} \sim R^3_{in}$, $R^2_{in}$, or $R_{in}$. Projecting the density of packaged polymer segments onto angular coordinates (*Figure 5—figure supplement 2*) reveals that the polymer is not homogenously distributed throughout the capsid surface, but instead has enriched density at the vertices and edges relative to the subunit faces. This result is consistent with experimental observations that nucleic acids form dodecahedral cages in viral particles (*Speir and Johnson, 2012*), and our model may describe scaling of the optimal charge ratio with volume for these capsids. For model capsids with $R_{in} \geq 12.5$ nm, the amount of polymer segments directly interacting with ARM charges becomes independent of capsid size, and the dependence of optimal length on volume can be attributed to the lengths of polymer between ARMs (see 'Discussion').

## Base-pairing increases packaged lengths

To understand how the geometric effects of base-pairing contribute to packaging, we performed dynamical assembly simulations and equilibrium calculations of $L^*_{eq}$ for a wide range of base-pairing patterns and fraction of base-paired nucleotides (see section 'Base-paired polymer'). The key result is that for all simulated base-pairing patterns, increasing the fraction of base-paired nucleotides (up to the biological fraction of 50%) increases $L^*_{eq}$ (*Figures 3C and 6D*). The increase in optimal length can be as large as 200–250 nucleotides for our small $T = 1$ capsid, indicating that base-pairing can contribute significantly to the amount of polymer that can be packaged. This effect can be explained by the fact that nucleotide–nucleotide interactions which drive NA structure formation effectively cancel some NA charge–charge repulsions and result in NA structures that are compact in comparison to linear polymers with the same lengths. Thus encapsulated NAs incur smaller excluded–volume interactions, electrostatic repulsions, and conformational entropy penalties during assembly.

However, the connection between the size of a molecule in solution and $L^*_{eq}$ is surprisingly subtle. As described in the section 'Base-paired polymer', we have quantified base-pairing patterns by their maximum ladder distance (MLD), which counts the maximum number of base-pairs along any non-repeating path across the NA and thus describes the extent of the molecule in the secondary structure graph space. As shown in *Figure 6*, for a NA with 1000 segments and 50% base-pairing, the solution radius of gyration varies with MLD as $R_G \sim \text{MLD}^{0.43}$ to yield $R_G \approx 8$ nm to $R_G \approx 20$ nm, in comparison the linear model $R_G = 25.5$ nm. As shown in *Figure 3C* the inclusion of base-pairing has a large effect on $L^*_{eq}$, but changes in MLD have only a minor effect. Though over this range of MLDs the solution $R_G$ more than doubles, $L^*_{eq}$ changes by only about 10%, with an even smaller variation over the range of MLDs that we estimate for biological RNA molecules MLD/Max MLD $\in (0.25, 0.55)$ based on *Gopal et al. (2012)* (see section 'Base-paired polymer' for additional detail).

## Semiflexible polymer

The effect of persistence length without tertiary structure (i.e., dsDNA) is shown in *Figure 3—figure supplement 1*.

## Effect of salt concentration

To understand the dependence of $L^*_{eq}$ on ionic strength and to evaluate the effect of the approximations made in the Debye–Huckel treatment of electrostatics, we performed a number of simulations using the primitive model representation of electrostatics and explicit ions to represent neutralizing counterions and added salt (the 'Model potentials and parameters' section). Ions are modeled as repulsive spheres (*Equation 6* below) and electrostatics are calculated according to Coulomb interactions (*Equation 12* below) with the relative permittivity set to 80.

As shown in *Figure 4*, the optimal length $L^*_{eq}$ and charge ratio increase with increasing ionic strength (i.e., decreasing Debye length $\lambda_D$). This effect can be explained by the fact that a smaller fraction of NA charges interact with positive capsid charges as the screening length decreases (see the 'Discussion' section). Importantly, the simulations predict overcharging at all salt concentrations investigated (1 mM $\leq C_{salt} \lesssim 400$ mM). Over this range, we see that optimal lengths predicted by simulations using explicit ions or Debye–Huckel interactions agree to within about 10% (*Figure 4*). The Debye–Huckel simulations slightly overpredict the optimal length at high salt concentrations because they neglect counterion excluded-volume, while they underpredict the optimal length at low ionic strength because they neglect ion–ion correlations. We also present the results of the limiting case where only neutralizing counterions are used (resulting in ~1 mM cations and 0 anions, for a total ionic strength of ~0.5 mM). Further simulations exploring the effect of divalent cations show only a slight increase in $L^*_{eq}$ at physiologically relevant divalent cation concentrations (1 mM) (*Figure 4*). Results of additional

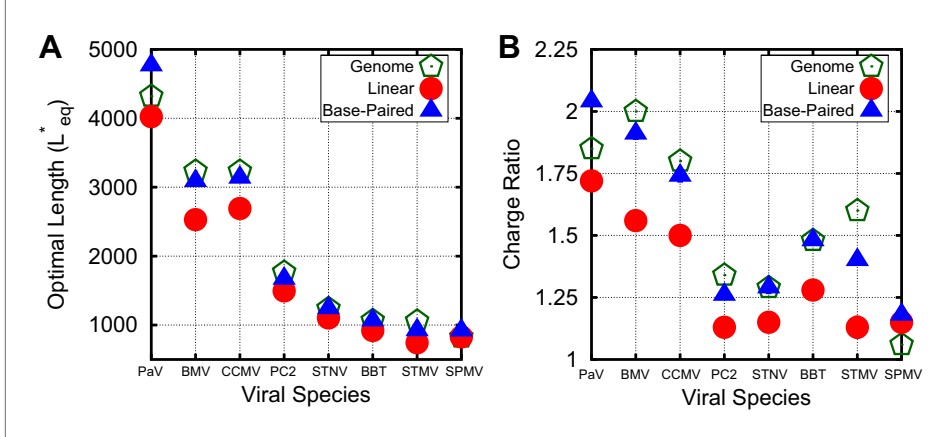

**Figure 4**. Correspondance between viral genomes and model calculations. Comparison between viral genome lengths and calculated thermodynamic optimal lengths (**A**) and charge ratios (**B**) for models based on the indicated viral capsid structures (see **Table 1**). Predicted optimal lengths are shown for linear polyelectrolytes (red circles) and model NAs (blue triangles) with 50% base-pairing. Viral genome lengths are shown with green pentagons. Error bars fall within the symbol sizes.

The following figure supplements are available for figure 4:

**Figure supplement 1**. Our capsid model can be modified to describe specific viral capsids by altering the capsid radius and ARM sequence.

**Figure supplement 2**. Optimal lengths are sensitive to multiple factors.

simulations examining the effect of ion size and charge renormalization are shown in *Figure 3—figure supplement 2* and *Figure 3—figure supplement 3*. We focus on $C_{salt}$ = 100 mM 1:1 salt for all other results in this article.

## Predictions for specific viral capsid structures

To evaluate the significance of the trends identified above for packaging in a biological context, we performed equilibrium calculations in which the structural parameters discussed above (capsid volume, ARM length, charge, and NA base-pairing) were based on specific $T = 1$ and $T = 3$ viruses (whose capsids are assembled from 60 and 180 protein copies respectively). For each investigated virus, the capsid radius was fit to protein densities in capsid crystal structures (*Carrillo-Tripp et al., 2009*), the ARM length was determined from the structure, and charges in the ARM and on the capsid inner surface were assigned based on amino acid sequence (see *Table 1*). NAs were modeled with 50% base-pairing and MLD/Max MLD ≈ 0.5. Visualizations of $T = 1$ and $T = 3$ viruses (PC2 and CCMV) are presented in *Figure 1D* and further details details are provided in *Figure 4—figure supplement 2*.

The predicted values of $L_{eq}^*$ for linear polyelectrolytes and base-paired NAs are compared to the actual viral genome lengths in *Figure 4*. We see that overcharging (charge ratios larger than 1, *Figure 4B*) is predicted for all structures. Furthermore, while the values of $L_{eq}^*$ predicted for linear polyelectrolytes fall short of the viral genome lengths for all investigated structures except for SPMV (whose virion has an unusually low charge ratio), $L_{eq}^*$ for the NA models are relatively close to the viral genome lengths for most structures. We emphasize that the optimal length is sensitive to all of the control parameters; for example, $L_{eq}^*$ is correlated not just with the capsid charge, but also with capsid volume and ARM packing fraction (see *Figure 4—figure supplement 2*). Recalling that $L_{eq}^*$ sets an upper bound on length of a polymer that can be efficiently packaged during assembly (*Figure 2B*), this result suggests that the geometric effects of base-pairing contribute to spontaneous packaging of viral genomes. The largest difference between $L_{eq}^*$ and genome length occurs for STMV. This discrepancy may reflect the fact that we used a NA base-pairing fraction of $f_{bp} = 0.5$ whereas 57% of nucleotides participate in secondary structure elements in the STMV crystal structure (*Larson et al., 1998*; *Zeng et al., 2012*) (lower fractions of nucleotides are resolved in other virion structures, suggesting lower values of $f_{bp}$).

**Table 1.** Details for the models of biological capsids studied in this article. The capsid inradius (distance from capsid center to face center), number of residues in the arginine rich motif (ARM), and net charge of the ARM and inner surface are features used to build these models. The viral genome length is then presented for comparison to the value of $L^*_{eq}$ predicted for the base-paired model. Finally, the fraction of occupied volume within the capsid is given for the base-paired model at the optimal length

| Virus | Inradius (nm) | ARM Length/Net charge | Genome length | $L^*_{eq}$ | Occupied volume fraction |
|-------|---------------|-----------------------|---------------|------------|--------------------------|
| PaV | 13.0 | 48/+13 | 4322 | 4766 | 0.074 |
| CCMV | 11.5 | 48/+10 | 3233 | 3136 | 0.099 |
| BMV | 11.5 | 44/+9 | 3233 | 3087 | 0.093 |
| PC2 | 8.0 | 43/+22 | 1767 | 1672 | 0.265 |
| STNV | 7.7 | 28/+16 | 1239 | 1242 | 0.240 |
| BBT | 7.5 | 27/+12 | 1066 | 1058 | 0.209 |
| STMV | 7.2 | 19/+11 | 1058 | 922 | 0.232 |
| SPMV | 6.8 | 20/+13 | 826 | 918 | 0.276 |

## Discussion

We have shown that assembly simulations and equilibrium calculations based on our coarse-grained model predict optimal NA lengths which are overcharged and relatively close to actual genome sizes for a number of viruses. This finding contrasts with earlier continuum models solved under an assumption of spherical symmetry, which required either a Donnan potential (*Ting et al., 2011*; *Ni et al., 2012*) or irreversible charge renormalization of the NA (*Belyi and Muthukumar, 2006*; see *Figure 3—figure supplement 3*) to account for overcharging. Our results (*Figures 2, 3, 4*) show that the optimal genome length is determined by a complex interplay between capsid charge, capsid size, excluded-volume, and RNA structure.

### The origins of overcharging

Analysis of conformations of encapsulated polymers in our simulations shows that overcharging arises as a consequence of geometry and electrostatic screening. The presence of discrete positive charges located in ARMs (or on the capsid surface) combined with nm-scale screening of electrostatics limits the number of direct NA–protein electrostatic interactions; the remaining nucleotides are found in segments which bridge the gaps between positive charges. These interconnecting (bridging) segments are the primary origin of overcharging. Earlier approaches which assumed spherical symmetry could not capture these bridging segments and thus did not predict overcharging. The significance of bridging segments to overcharging is clearly revealed by the dependence of optimal length on capsid size under constant ARM length (*Figures 3B*). For $R_{in} \geq 12.5$ nm, the amount of NA interacting with the ARMs is constant, while bridging lengths increase with capsid radius (*Figure 5—figure supplement 3*) due to the increased typical distance between charges on different ARMs. The increase in $L^*_{eq}$ with capsid radius in these calculations can be attributed entirely to increased bridge lengths.

Although the amounts of bridging segments in the biological capsid models depend on many control parameters (e.g., charge, volume, packing fraction, RNA structure), we also confirmed the significance of bridging segments to overcharging in these calculations. *Figure 5* breaks down the $L^*_{eq}$ into the number of segments which interact with positive ARM charges and the number of segments which are bridging. If one counts only the NA segments that directly interact with capsid charges, the resulting charge ratio is slightly less than one for each of these capsids. However, when the bridging segments are included, all the capsids are overcharged. Interestingly, more bridging segments are found in the larger $T = 3$ capsids (56% bridging) than in $T = 1$ capsids (25% bridging), contributing to the larger predicted charge ratios for $T = 3$ capsids (*Figure 4B*). Though the fraction of nucleotides closely interacting with protein in capsids is difficult to measure experimentally, it might be estimated from the amounts of RNA resolved in crystallographic or EM structures. In a recent summary, Larsson et al. found that for 10 $T = 3$ crystal structures an average of 16% of NA were resolved. For $T = 1$ structures a wider

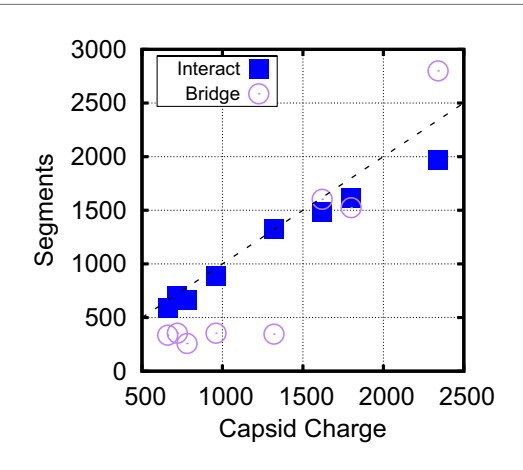

**Figure 5**. Bridging in biological capsids. Number of NA segments that directly interact with positively charged ARM segments (interaction energy $\leq -0.5k_BT$, blue squares) and bridging segments (interaction energy $> -0.5k_BT$, purple circles). The numbers are calculated at the optimal length $L^*_{eq}$ for each capsid shown in **Figure 4** using the base-paired model. For visual reference, the dashed line indicates a 1:1 correspondence between capsid charge and number of nucleotides.

The following figure supplements are available for figure 5:

**Figure supplement 1**. Radial density for linear polymer and ARM segments in the simple capsid (A) and CCMV (B).

**Figure supplement 2**. Angular density of linear polymer segments (heat map) in the basic capsid model (A) and CCMV (B).

**Figure supplement 3**. Capsid radius and polymer bridging.

**Figure supplement 4**. Number of NA segments that directly interact with positively charged ARM segments and bridging segments, for both the linear and base-paired model.

range of values was obtained, where some had a large fraction of NA resolved (STMV = 62%, STNV = 34%), but other ssDNA viruses had resolved fractions similar to $T = 3$ viruses. An additional piece of evidence comes from low resolution neutron diffraction, where 72% of RNA was observed to be in the first layer of density along the inner capsid surface of the $T = 1$ STNV, again suggesting that much of the $T = 1$ viral genome is closely interacting with the protein (***Bentley et al., 1987***). We present additional data describing the conformation of the polymer within the capsid, including the radial and angular densities as ***Figure 5— figure supplements 1, 2, 3, 4***.

We emphasize that our coarse-grained model is designed to incorporate the minimal set of features required to explain the thermodynamic stability of viral particles, and thus neglects some factors that contribute to packaging specific NAs. The in vivo experiments in ***Ni et al. (2012)*** on brome mosaic virus (BMV) showed that even charge-conserving mutations to ARM residues could affect the amounts and types of packaged RNA, possibly by interfering with coordination of RNA replication and encapsidation (***Rao, 2006***). Similarly, packaging signals, or regions of RNA that have sequence-specific interactions with the capsid protein, are known for some viruses (e.g., HIV [***D'Souza and Summers, 2005***] or MS2 and satellite tobacco necrosis virus [STNV] [***Bunka et al., 2011***; ***Borodavka et al., 2012***]). Packaging signals could be added to our model to investigate how they favor selective assembly around the viral genome through kinetic (***Borodavka et al., 2012***) or thermodynamic effects. The fact that our model predicts $L^*_{eq}$ for STNV close to its genome length without accounting for sequence specificity may suggest that packaging signals have only a small effect on the thermodynamic $L^*_{eq}$.

In conclusion, our results elucidate the connection between structure and assembly for viral capsids. Firstly, our simulations show that 'overcharged' capsids are favored thermodynamically and kinetically, even in the absence of cellular factors or other external effects. Secondly, our results delineate how the genome length which is most favorable for assembly depends on virus-specific quantities such as capsid charge, capsid volume, and genomic tertiary structure. While the correlation between predicted optimal lengths and viral genome sizes suggests that our results have biological relevance, the physical foundations of our model can be tested via controlled in vitro experiments. As noted above, several existing experimental observations agree with our results. A positive correlation between protein charge and amounts of packaged RNA has been demonstrated through experiments in which the charge on capsid protein ARMs was altered by mutagenesis (e.g., [***Dong et al., 1998***; ***Kaplan et al., 1998***; ***Venter et al., 2009***]). Competition assays (***Porterfield et al., 2010***; ***Comas-Garcia et al., 2012***), in which two species of NAs or other polymers compete for packaging by a limiting concentration of capsid proteins, offer a quantitative estimate of $L^*_{eq}$. For example, our prediction that $L^*_{eq}$ for CCMV is roughly consistent with the genome length (***Figure 4***) agrees with the observation that CCMV proteins preferentially package longer RNAs, up to the

wildtype genome length, over shorter RNAs in competition assays (**Comas-Garcia et al., 2012**). Now, it is possible to quantitatively test the predictions of our model for the dependence of $L_{eq}^*$ on protein charge and salt concentration through similar competition assays in which NA length preferences are observed for proteins with charge altered by mutagenesis under different ionic strengths. Similarly, our prediction that base-pairing increases $L_{eq}^*$ can be evaluated by comparison of assembly experiments on RNA and synthetic polyelectrolytes (e.g., polystyrene sulfonate) or RNA with base-pairing inhibited through chemical modification (e.g., etheno-RNA [**Dhason et al., 2012**]). Our simulations predict that above the optimal length for a linear polyelectrolyte, only base-paired RNA will be packaged in high yields of well-formed capsids.

## Methods

### Model description

We have extended a model for empty capsid assembly (**Wales, 2005**; **Fejer et al., 2009**; **Johnston et al., 2010**) to describe assembly around NAs. A complete listing of the interaction potentials is provided below; here we present a concise description of our model. The pseudoatoms in the capsid subunit model are illustrated in **Figure 1**. Subunit assembly is mediated through an attractive Morse potential between Attractor ('A') pseudoatoms located at each subunit vertex. The Top ('T') pseudoatoms interact with other 'T' psuedoatoms through a potential consisting of the repulsive term of the LJ potential, the radius of which is chosen to favor a subunit-subunit angle consistent with a dodecahedron (116°). The Bottom ('B') pseudoatom has a repulsive LJ interaction with 'T' pseudoatoms, intended to prevent 'upside-down' assembly. The 'T', 'B', and 'A' pseudoatoms form a rigid body (**Wales, 2005**; **Fejer et al., 2009**; **Johnston et al., 2010**). See **Schwartz et al. (1998)**, **Rapaport et al. (1999)**, **Rapaport (2004**, **2008)**, **Hagan and Chandler (2006)**, **Hicks and Henley (2006)**, **Nguyen et al. (2007)**, **Wilber et al., (2007**, **2009a**, **2009b)**, **Hagan (2008)**, **Nguyen and Brooks (2008)**, **Nguyen et al. (2009)**, **Elrad and Hagan (2010)**, **Johnston et al. (2010)**, **Hagan et al. (2011)**, **Mahalik and Muthukumar (2012)**, **Hagan (2013)** for related models.

To model electrostatic interaction with a negatively charged NA or polyelectrolyte we extend the model as follows. Firstly, to better represent the capsid shell we add a layer of 'Excluder' pseudoatoms which have a repulsive LJ interaction with the polyelectrolyte and the ARMs. Each ARM is modeled as a bead–spring polymer, with one bead per amino acid. The 'Excluders' and first ARM segment are part of the subunit rigid body. ARM beads interact through repulsive Lennard–Jones interactions and, if charged, electrostatic interactions modeled by a Debye–Huckel potential. Comparison to Coulomb interactions with explicit counterions is shown in **Figure 3D**. We also show that the results do not change significantly when experimentally relevant concentrations of divalent cations are added to the system (**Figure 3D**). The ability of the Debye–Huckel model to provide a reasonable representation of electrostatics in the system can be understood based on the relatively low packing fractions (see **Table 1**) within the assembled capsids and the fact that the relevant experimental and physiological conditions correspond to moderate to high salt concentrations.

Simulations were performed with the Brownian Dynamics algorithm of HOOMD, which uses the Langevin equation to evolve positions and rigid body orientations in time (**Anderson et al., 2008**; **Nguyen et al., 2011**; **LeBard et al., 2012**). Simulations were run using a set of fundamental units. The fundamental energy unit is selected to be $E_u \equiv 1 k_B T$. The unit of length $D_u$ is set to the circumradius of a pentagonal subunit, which is taken to be $1 D_u \equiv 5$ nm so that the dodecahedron inradius of $1.46 D_u = 7.3$ nm gives an interior volume consistent with that of the smallest $T = 1$ capsids. Assembly simulations were run at least 10 times for each set of parameters, each of which were concluded at $2 \times 10^8$ time steps. The following parameters values were used in all of our dynamical assembly simulations: $\lambda_D = 1$ nm, box size $= 200 \times 200 \times 200$ nm, subunit concentration $= 12 \mu M$. During calculation of the thermodynamic optimal polymer length $L_{eq}^*$, calculations were run at least $1 \times 10^7$ timesteps, with equilibrium assessed after convergence. Standard error was obtained based on averages of multiple ($\geq 3$) independent simulations. Separate calculations of $L_{eq}^*$ were also performed using using the Widom test-particle method (**Widom, 1963**) as extended to calculate polymer residual chemical potentials (**Kumar et al., 1991**; **Elrad and Hagan, 2010**) (described in more detail below). Snapshots from simulations were visualized using VMD (**Humphrey et al., 1996**).

## Base-paired polymer

To obtain base-paired polymers with a wide and tunable range of structures (i.e., maximum ladder distances), we implement the following strategy. Firstly, the polymer contour length $L_C$, length of the base-paired segments $L_{bp}$, and fraction of nucleotides in base-pairs $f_{bp}$ are free parameters which we specify (typical values are $L_C = 1000$ nucleotides, $L_{bp} = 5$ nucleotides per segment, and $f_{bp} = 0.5$). Secondly, we iterate over the linear sequence of the polymer, randomly choosing segments which will undergo base-pairing to form double-stranded (ds) segments. Each segment consists of $L_{bp}$ consecutive nucleotides. Segments are numbered sequentially to facilitate pairing (i.e., the first ds segment in the sequence is 1, the second is 2, and so on). Thirdly, these ds segments are then paired together. In the case of the hairpin model, each ds strand is paired with the next ds segment in the sequence (i.e., the first segment with the second, third with fourth, and so on). In the general base-pairing model, pairs are assigned stochastically according to an algorithm which allows us to simultaneously tune the distribution of junction orders and the maximum ladder distance (MLD). The algorithm is described in *Figure 6A* defined as follows:

The first step in assigning a base-pair is to obtain a random separation $l_{random}$ from an exponential distribution where $\lambda$ is the inverse of the mean:

$$\left( l_{random}\left( \lambda, l \right) \right) = \lambda e^{-l\lambda}. \qquad (1)$$

To prevent pseudoknots this $l_{random}$ is then subtracted from the maximal available separation $l_{max}$ to yield $l_{pair}$:

$$l_{pair}\left( l_{max}, l_{random} \right) = l_{max} - l_{random}. \qquad (2)$$

The obtained $l_{pair}$ defines the number of segments separating the current segment from its base-pairing partner. With this algorithm, the single control parameter parameter $\lambda$ is used to control both the base-pairing pattern, and thus MLD and the distribution of junction types, that is, the number of double stranded segments emerging from a single stranded intersection (see *Figure 6C*). When $\lambda$ is large, we are more likely to obtain small values of $l_{random}$, and thus large values of $l_{pair}$. Large $l_{pair}$ values lead to more extensive structures (i.e., larger MLDs and a larger fraction of two-junctions). When $\lambda$ is lower, we have a broader distribution of $l_{random}$ values, and thus obtain smaller values of $l_{pair}$. If $l_{pair}$ is small, it creates higher-order junctions and regions which are not part of the MLD.

To describe the structures of the polymers generated by this algorithm, we make use of two structural parameters: the maximum ladder distance (MLD) and radius of gyration ($R_G$). As in (*Yoffe et al., 2008*), we define the MLD as the largest number of base-pairs in any single path across the molecule's secondary structure. *Figure 6B* describes the polymer radius of gyration $R_G$ as a function of MLD, normalized by the maximal possible MLD (i.e., if all base-pairs were along a single path), for polymers of length 1000 with fraction base-pairing $f_{bp} = 0.5$. All of the base-paired polymers are compressed relative to the linear polymer ($R_G = 25.5$ nm), but they differ amongst themselves significantly. We observe $R_G$ to vary with MLD as $R_G \sim MLD^{0.43}$ to yield sizes in the range $R_G \approx 8$ nm to $R_G \approx 20$.

## Effect of MLD on optimal charge ratio

In order to estimate biological MLD values, we fit the histogram of junction numbers generated by our algorithm with different values of $\lambda$ and against the distribution of junction numbers obtained for biological ssRNA molecules in *Gopal et al. (2012)* (*Figure 6C*). For the two cellular, noncoding ssRNA segments, we obtain normalized MLDs of 0.55 and 0.36, and for a viral segment (RNA2 of CCMV) we obtain 0.25. As shown in *Figure 6B* the radii of gyration for RNAs with lengths of $L_C = 1000$ nt and the normalized MLDs of the cellular RNAs of 0.55 and 0.36 are respectively 14.1 nm and 11.8 nm. A 1000 nt RNA with the viral normalized MLD of 0.25 has $R_G = 10.1$ nm; that is, the viral-like RNA is compressed by 14–29% in solution. However, as shown in *Figure 3C*, the optimal charge ratios for these RNAs in the simple capsid model are within the large statistical error (we obtain 2.70, 2.75, and 2.78 respectively from a linear fit to the data). An example assembly simulation is shown in *Figure 6E* and *Video 2*.

## Subunit–subunit binding free energy estimates

Our method of calculating the subunit–subunit binding free energy is similar to that presented in our previous simulations (*Elrad and Hagan, 2010*; *Hagan et al., 2011*). Briefly, subunits were modified

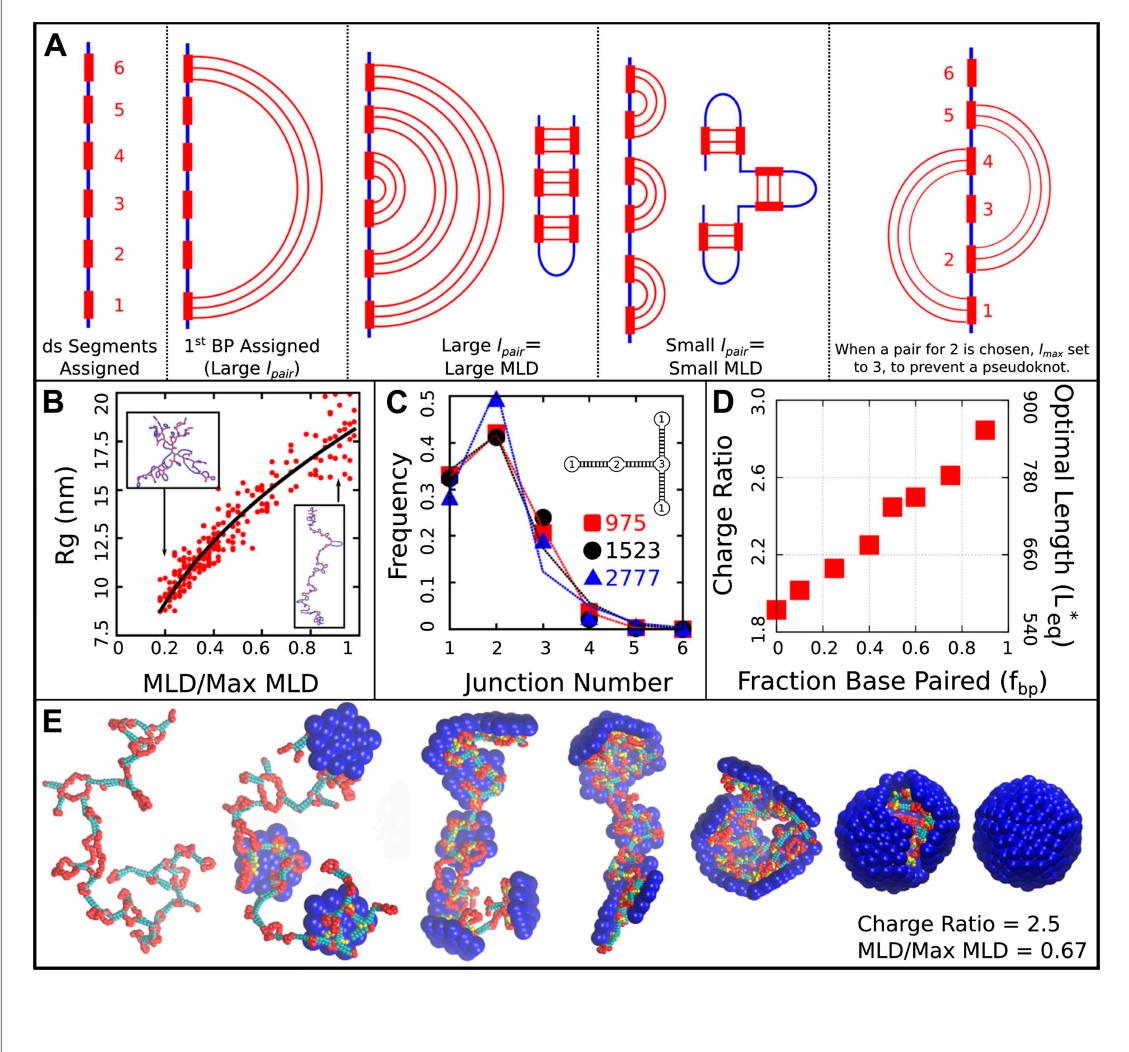

**Figure 6**. Base-paired polymer setup and analysis. (**A**) Schematic illustrations of the algorithm we used to obtain a wide range of base-paired structures. From left to right, double-stranded (ds) segments are first randomly assigned. These segments are then base-paired together, starting from one end. If base-paired segments are widely separated (i.e., $l_{pair}$ is large) then subsequent nested base-pairs lead to an extended structure. Conversely, if $l_{pair}$ is small, less extended structures form. The right-most panel indicates a psuedoknot, a structural motif we have prevented from occurring in this model, by setting $l_{max}$ to the last unpaired segment. (**B**) Radius of gyration $R_G$ for model NAs isolated in solution as a function of maximum ladder distance (MLD) normalized by the maximum possible MLD. The nucleic acid has 1000 nt, 50% of which are base-paired. (**C**) The frequency of junction numbers can be altered by varying $\lambda$ in **Equation 2**, with large values of $\lambda$ leading to large values of $l_{pair}$. The symbols indicate the relative frequency of junction numbers for biological RNAs with indicated lengths, obtained from Ref. (**Gopal et al., 2012**), and the lines are best fits to these distributions generated by varying $\lambda$. The inset illustrates several different junction orders. (**D**) The thermodynamic optimum length measured for the simple model capsid as a function of the fraction of base-paired nucleotides $f_{bp}$ for a simplified 'hairpins only' model (red squares). (**E**) Snapshots illustrating assembly around a NA. Beads are colored as follows: blue = excluders, yellow = ARM bead, red = single-stranded NA, cyan = double-stranded NA. 'Top', 'Bottom', and 'Attractor' beads removed for clarity.

such that only one edge formed attractive bonds, limiting complex formation to dimers. We measured the relative concentration of dimers and monomers for a range of attraction strengths ($\varepsilon$). The free energy of binding along that interface is then $g_{cc} = -k_B T \ln(c_{ss}/K_d)$ with standard state concentration $c_{ss} = 1\,\text{M}$ and $K_d$ in molar units. This free energy is well fit by the linear expression $g_{cc} - 1.5\varepsilon - Ts_b$ where $s_b = -9k_B T$. We can then correct for the multiplicity of dimer conformations, by adding in the additional term $T - \Delta s_c = \ln(25/2)$, where the five pentagonal edges are assumed to be distinguishable, but complex orientations which differ only through global rotation are not. Our assembly simulations are run at $\varepsilon = 5k_B T$, for which we observe only transient subunit–subunit associations except in the

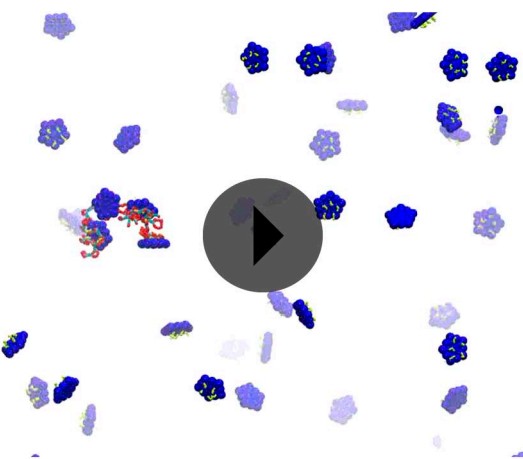

**Video 2.** Base-Paired Capsid Assembly. Movie illustrating assembly of subunits with ARM length=5 around a base-paired molecule with a charge ratio of 2.5 and normalized MLD of 0.67. Beads are colored as in *Figure 6*.

presence of an anionic polyelectrolyte. Our free energy calculations agree with this observation, suggesting that for this value of $\varepsilon$ binding is very weak: $K_d = 0.33M$ and $g_{cc} = -1.1k_BT$. Note that formation of additional bonds in a capsid structure will give rise to substantially more negative binding free energies. As shown in *Hagan and Chandler (2006)* much of the binding entropy penalty associated with adding a subunit to a capsid is incurred during the formation of the first bond, with smaller decreases in entropy associated with forming additional bonds. A similar set of calculations for capsids with the ARMs removed decreased the binding free energy to $g_{cc} = -1.84k_BT$, indicating that ARM–ARM interactions increase the free energy by about $0.74k_BT$ along each interface at $C_{salt} = 100$ mM.

## Equilibrium encapsidation

The free energy as a function of encapsidated polymer length was obtained by two different methods. In the first, we placed a very long polymer in or near a preassembled capsid, with one of the capsid subunits made permeable to the polymer. We then performed unbiased Brownian dynamics. Once the amount of packaged polymer reached equilibrium, the thermodynamic optimum length $L^*_{eq}$ and the distribution of fluctuations around it were measured.

In the second approach we used the Widom test-particle method (*Widom, 1963*) as extended to calculate polymer residual chemical potentials (*Kumar et al., 1991*; *Elrad and Hagan, 2010*). We performed independent sets of simulations for a free and an encapsidated polymer in which we calculated the residual chemical potential $\mu_r$ according to:

$$-\beta\mu_r\left(N_p\right) \equiv -\beta\left(\mu_{chain}\left(N_p+1\right)-\mu_{chain}\left(N_p\right)\right)$$
$$= \log\left\langle\exp\left(-\beta U_I\left(N_p\right)\right)\right\rangle$$

(3)

where $N_p$ is the number of segments in the polymer and $U_I$ is the interaction energy experienced by a test segment inserted onto either end of the polymer. Importance sampling was used to make the calculation feasible, where the bond length of inserted segments was chosen from a normal distribution matching the equilibrium distribution of bond lengths, truncated at ±3 standard deviations. The effect of using this biased insertion was removed a posteriori according to standard non-Boltzmann sampling. Between incrementing $N_p$, $10^5$ steps of dynamics were run, and $10^5$ insertions were attempted for each value of $N_p$ in 100 independent runs. The results of these calculations are presented in *Figure 2—figure supplement 2*, and based on the point of intersection between the encapsidated and unencapsidated chemical potentials, we estimate the optimal length $L^*_{eq}$ to be between 550–575 segments (or a charge ratio of $1.83-1.92$), which is close agreement with the preassembled dynamics calculations (574 segments or a charge ratio of 1.91). If we integrate the difference in chemical potential between the encapsidated and unencapsidated polymers between 0 and 575, we obtain $-500kT$ as an estimate for the free energy of polymer encapsidation due to both polymer–ARM and polymer–polymer interactions in the simple capsid model with ARM length = 5. Since the primary motivation for the Widom test-particle method calculations was to provide an independent test of optimal lengths calculated using the semipermeable capsid, we only considered the Debye–Huckel model for electrostatics in test-particle method calculations.

To further assess the convergence and sampling of both approaches for calculating the $L^*_{eq}$, we performed additional replica exchange (REX) simulations (*Sugita and Okamoto, 1999*). In replica exchange, replicas of the system are simulated in parallel at different temperatures. Periodically, structures are exchanged between temperatures based on the Metropolis Criterion. In our systems,

12 replicas were run, with temperatures distributed exponentially between 1 kT and 1.5 kT. This resulted in a satisfactory exchange frequency of 30–40%. We present the results for REX simulations in *Figure 2—figure supplement 2*, but in this case and all other cases, the REX results quantitatively agree with the results of our simulations run at a single temperature.

## Model potentials and parameters

In our model, all potentials can be decomposed into pairwise interactions. Potentials involving capsomer subunits further decompose into pairwise interactions between their constituent building blocks—the excluders, attractors, 'Top' and 'Bottom', and ARM pseudoatoms. It is convenient to write the total energy of the system as the sum of 6 terms: a capsomer-capsomer $U_{cc}$ part (which does not include interactions between ARM pseudoatoms), capsomer-polymer $U_{cp}$, capsomer-ARM $U_{ca}$, polymer-polymer $U_{pp}$, polymer-ARM $U_{pa}$, and ARM-ARM $U_{aa}$ parts. Each is summed over all pairs of the appropriate type:

$$U = \sum_{\text{cap } i} \sum_{\text{cap } j<i} U_{cc} + \sum_{\text{cap } i} \sum_{\text{poly } j} U_{cp} + \sum_{\text{cap } i} \sum_{\text{ARM } j} U_{ca} +$$
$$\sum_{\text{poly } i} \sum_{\text{poly } j<i} U_{pp} + \sum_{\text{poly } i} \sum_{\text{ARM } j} U_{pa} + \sum_{\text{ARM } i} \sum_{\text{ARM } j<i} U_{aa} \tag{4}$$

where $\sum_{\text{cap } i} \sum_{\text{cap } j<i}$ is the sum over all distinct pairs of capsomers in the system, $\sum_{\text{cap } i} \sum_{\text{poly } j}$ is the sum over all capsomer-polymer pairs, etc.

The capsomer-capsomer potential $U_{cc}$ is the sum of the attractive interactions between complementary attractors, and geometry guiding repulsive interactions between 'Top'–'Top' pairs and 'Top'–'Bottom' pairs. There are no interactions between members of the same rigid body, but ARMs are not rigid and thus there are intra-subunit ARM–ARM interactions. Thus, for notational clarity, we index rigid bodies and non-rigid pseudoatoms in Roman, while the pseudoatoms comprising a particular rigid body are indexed in Greek. For example, for capsomer $i$ we denote its attractor positions as $\{a_{i\alpha}\}$ with the set comprising all attractors $\alpha$, its 'Top' positions $\{t_{i\alpha}\}$, and its 'Bottom' positions $\{b_{i\alpha}\}$. The capsomer–capsomer interaction potential between two capsomers $i$ and $j$ is then defined as:

$$U_{cc}(\{a_{i\alpha}\},\{t_{i\alpha}\},\{b_{i\alpha}\},\{a_{j\beta}\},\{t_{j\beta}\},\{b_{j\beta}\}) = \sum_{\alpha,\beta}^{N_t} \varepsilon\mathcal{L}\left(\left|t_{i\alpha} - t_{j\beta}\right|,\sigma_t\right) + \sum_{\alpha,\beta}^{N_b,N_t} \varepsilon\mathcal{L}\left(\left|b_{i\alpha} - t_{j\beta}\right|,\sigma_b\right) +$$
$$\sum_{\alpha,\beta}^{N_a} \varepsilon\mathcal{M}\left(\left|a_{i\alpha} - a_{j\beta}\right|,r_0,\varrho,r_{cut}\right) \tag{5}$$

where $\varepsilon$ is an adjustable parameter which both sets the strength of the capsomer–capsomer attraction at each attractor site and scales the repulsive interactions which enforce the dodecahedral geometry. $N_t$, $N_b$, and $N_a$ are the number of 'Top', 'Bottom', and attractor pseudoatoms respectively in one capsomer, $\sigma_t$ and $\sigma_b$ are the effective diameters of the 'Top'–'Top' interaction and 'Bottom'–'Top' interactions, which are set to 10.5 nm and 9.0 nm respectively throughout this work, $r_0$ is the minimum energy attractor distance, set to 1 nm, $\varrho$ is a parameter determining the width of the attractive interaction, set to 2.5, and $r_{cut}$ is the cutoff distance for the attractor potential, set to 10.0 nm.

The function $\mathcal{L}$ is defined as the repulsive component of the Lennard–Jones potential shifted to zero at the interaction diameter:

$$\mathcal{L}(x,\sigma) \equiv \begin{cases} \left(\dfrac{\sigma}{x}\right)^{12} - 1 & : x < \sigma \\ 0 & : \text{otherwise} \end{cases} \tag{6}$$

The function $\mathcal{M}$ is a Morse potential:

$$\mathcal{M}(x,r_0,\varrho) \equiv \begin{cases} \left(e^{\varrho\left(1-\frac{x}{r_0}\right)} - 2\right)e^{\varrho\left(1-\frac{x}{r_0}\right)} & : x < r_{cut} \\ 0 & : \text{otherwise} \end{cases} \tag{7}$$

The capsomer–polymer interaction is a short-range repulsion that accounts for excluded-volume. For capsomer $i$ with excluder positions $\{x_{i\alpha}\}$ and polymer subunit $j$ with position $R_j$, the potential is:

$$U_{cp}\left(\{\boldsymbol{x}_{i\alpha}\},\boldsymbol{R}_j\right)=\sum_{\alpha}^{N_x}\mathcal{L}\left(\left|\boldsymbol{x}_{i\alpha}-\boldsymbol{R}_j\right|,\sigma_{xp}\right) \tag{8}$$

where $N_x$ is the number of excluders on a capsomer and $\sigma_{xp}=0.5\left(\sigma_x+\sigma_p\right)$ is the effective diameter of the excluder–polymer repulsion. The diameter of the polymer bead is $\sigma_p=0.5$ nm and the diameter for the excluder beads is $\sigma_x=3.0$ nm for the $T=1$ model and $\sigma_x=5.25$ nm for the $T=3$ model.

The capsomer–ARM interaction is a short-range repulsion that accounts for excluded-volume. For capsomer $i$ with excluder positions $\{\boldsymbol{x}_{i\alpha}\}$ and ARM subunit $j$ with position $\boldsymbol{R}_j$, the potential is:

$$U_{cA}\left(\{\boldsymbol{x}_{i\alpha}\},\boldsymbol{R}_j\right)=\sum_{\alpha}^{N_x}\mathcal{L}\left(\left|\boldsymbol{x}_{i\alpha}-\boldsymbol{R}_j\right|,\sigma_{xA}\right) \tag{9}$$

with $\sigma_{xA}=0.5\left(\sigma_x+\sigma_A\right)$ as the effective diameter of the excluder–ARM repulsion with $\sigma_A=0.5$ nm the diameter of an ARM bead.

## Electrostatic interactions among polymer, ARM, and ion beads

The polymer–polymer non-bonded interaction is composed of electrostatic repulsions and short-ranged excluded-volume interactions. These polymers also contain bonded interactions which are only evaluated for segments occupying adjacent positions along the polymer chain and angular interactions which are only evaluated for three sequential polymer segments. As noted in the main text, electrostatics are represented either by Debye–Huckel interactions or by Coulomb interactions with explicit salt ions. For the case of Debye–Huckel interactions,

$$U_{pp}\left(\boldsymbol{R}_i,\boldsymbol{R}_j,\boldsymbol{R}_k\right)=\begin{cases}\mathcal{K}_{bond}\left(R_{ij},\sigma_p,k_{bond}\right) & :\{i,j\}\,\text{bonded}\\[4pt]\mathcal{K}_{angle}\left(R_{ijk},k_{angle}\right) & :\{i,j,k\}\,\text{angle}\\[4pt]\mathcal{L}\left(R_{ij},\sigma_p\right)+\mathcal{U}_{DH}\left(R_{ij},q_p,q_p,\sigma_p\right) & :\{i,j\}\,\text{nonbonded}\end{cases} \tag{10}$$

where $R_{ij}\equiv\left|\boldsymbol{R}_i-\boldsymbol{R}_j\right|$ is the center-to-center distance between the polymer subunits, $q_p=-1$ is the valence of charge on each polymer segment, and $\mathcal{U}_{DH}$ is a Debye–Huckel potential smoothly shifted to zero at the cutoff:

$$\mathcal{U}_{DH}\left(r,q_1,q_2,\sigma\right)\equiv\begin{cases}\dfrac{q_1q_2l_b\,e^{\sigma/\lambda_D}}{\lambda_D+\sigma}\left(\dfrac{e^{-r/\lambda_D}}{r}\right) & :x<2\lambda_D\\[10pt]\dfrac{(r_{cut}^2-r^2)^2(r_{cut}^2+2r^2-3r_{on}^2))}{\left(r_{cut}^2-2r_{on}^2\right)^3}\dfrac{q_1q_2l_b\,e^{\sigma/\lambda_D}}{\lambda_D+\sigma}\left(\dfrac{e^{-r/\lambda_D}}{r}\right) & :2\lambda_D<x<3\lambda_D\\[10pt]0 & :\text{otherwise}\end{cases} \tag{11}$$

$\lambda_D$ is the Debye length, $l_b$ is the Bjerrum length, and $q_1$ and $q_2$ are the valences of the interacting charges. For the cases using explicit electrostatics the $\mathcal{U}_{DH}$ potential is replaced by a Coulomb potential:

$$\mathcal{C}\left(r,q_1,q_2\right)\equiv\frac{q_1q_2}{4\pi\varepsilon_0\varepsilon_r}\frac{1}{r} \tag{12}$$

where $4\pi\varepsilon_0$ is the term for the permittivity of free space and $\varepsilon_r$ is the relative permittivity of the solution, set to 80. Above a cutoff distance ($r_{cut}=5$ nm) the electrostatics are calculated using the particle-particle particle-mesh (PPPM) Ewald summation (*LeBard et al., 2012*). Explicit ions are included in these simulations to represent both neutralizing counterions and added salt. Ions interact with other charged beads in the solution according to the Coulomb potential (*Equation 12*) and interact with all beads through the repulsive shifted LJ interaction (*Equation 6*). Except for the results in *Figure 3—figure supplement 2*, ionic radii were set to 0.125 nm (i.e., $\sigma=0.25$ nm in *Equation 6* below), which is roughly equal to the radii of Na$^+$ and CL$^-$ ions in the CHARMM force field (*Beglov and Roux, 1994*; *MacKerell et al., 1998*; *Mackerell, 2004*).

Specific binding between $Mg^{2+}$ ions and RNA is known to affect RNA structure. To test the effect of such stably bound divalent cations on optimal length, we constructed a polyelectrolyte with a divalent cation irreversible bound (through a harmonic potential, see *Equation 13*) to every 100th NA segment, in a solution containing 100 mM 1:1 salt. While this model does not capture the structural effects of specific $Mg^{2+}$ binding to RNA, it does represent the fact that these bound cations effectively cancel some NA charges.

Bonds are represented by a harmonic potential:

$$\mathcal{K}_{bond}\left(R_{ij}, \sigma, k_{bond}\right) \equiv \frac{k_{bond}}{2}\left(R_{ij} - \sigma\right)^2. \tag{13}$$

Angles are also represented by a harmonic potential:

$$\mathcal{K}_{angle}\left(R_{ijk}, k_{angle}\right) \equiv \frac{k_{angle}}{2}\left(\vartheta_{ijk}\right)^2 \tag{14}$$

where $\vartheta_{ijk}$ is the angle formed by the sequential polymer units $i, j, k$.

The ARM–ARM interaction is similar to the polymer–polymer interaction, consisting of non-bonded interactions composed of electrostatic repulsions and short-ranged excluded-volume interactions. These ARMs also contain bonded interactions which are only evaluated for segments occupying adjacent positions along the polymer chain:

$$U_{aa}\left(\boldsymbol{R}_i, \boldsymbol{R}_j\right) = \begin{cases} \mathcal{K}_{bond}\left(R_{ij}, \sigma_a, k_{bond}\right) & : \{i, j\}\,\text{bonded} \\ \mathcal{L}\left(R_{ij}, \sigma_a\right) + \mathcal{U}_{DH}\left(R_{ij}, q_i, q_j, \sigma_a\right) & : \{i, j\}\,\text{nonbonded} \end{cases} \tag{15}$$

where $R_{ij} \equiv \left|\boldsymbol{R}_i - \boldsymbol{R}_j\right|$ is the center-to-center distance between the ARM subunits and $q_i$ is the valence of charge on ARM segment $i$.

Finally, the ARM–Polymer interaction is the sum of short-ranged excluded-volume interactions and electrostatic interactions:

$$U_{pa}\left(\boldsymbol{R}_i, \boldsymbol{R}_j\right) = \mathcal{L}\left(R_{ij}, \sigma_{ap}\right) + \mathcal{U}_{DH}\left(R_{ij}, q_i, q_j, \sigma_{ap}\right) \tag{16}$$

## Acknowledgements

We gratefully acknowledge Chuck Knobler, William Gelbart, and Adam Zlotnick for insightful discussions and critical reads of the manuscript.

## Additional information

### Funding

| Funder | Grant reference number | Author |
| --- | --- | --- |
| National Institutes of Health, National Institute of Allergy and Infectious Diseases | R01AI080791 | Michael F Hagan |
| National Science Foundation XSEDE (Keeneland, Longhorn, and Condor) | TG-MCB090163 | Michael F Hagan |

The funders had no role in study design, data collection and interpretation, or the decision to submit the work for publication.

## Author contributions

JDP, CQ, MFH, Conception and design, Acquisition of data, Analysis and interpretation of data, Drafting or revising the article

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
