## [Decision Letter]

Thank you for sending your work entitled “Viral genome structures are optimal for capsid assembly” for consideration at *eLife*. Your article has been favorably evaluated by a Senior editor and 2 reviewers, one of whom is a member of our Board of Reviewing Editors.

The Reviewing editor and the other reviewer discussed their comments before we reached this decision, and the Reviewing editor has assembled the following comments to help you prepare a revised submission.

While both reviewers agree that the study is of very high interest, there is a need to provide further clarification and justification for the treatment of electrostatic screening in the model and its consequences on the robustness of the overall conclusions. The manuscript is clearly and cleanly written, logical, beautifully carried out, and of significance for the understanding of viral assembly and function. The authors first show dynamical assembly of capsids around NAs occurs optimally for parameters that thermodynamically optimize assembly. This allows for more controlled thermodynamic simulations that disentangle the role of charge, excluded volume, and tertiary structure have in the assembly process. Significantly, the authors find that assembled capsids spontaneously overcharge in the absence of eternal fields or influences. In all, one finds no flaws with the manuscript, either stylistically or in terms of the science. The study adds significantly to the field of virus biophysics. However, because the simulations are based on a simplified model of viral capsid assembly, there could be serious concerns about all the approximations that are made. Some clarification by the authors would be welcome to further strengthen the study.

1) Part of the appeal of the study is the simplicity of the underlying model, which confers great clarity to the analysis and the description of the results. The nicest and most impressive results are probably summarized in Figure 4. However, upon more scrutiny, one does also start to worry about the significance of the approximations that are being made, and how much confidence one has in the robustness of the conclusions. For example, it would be nice to convey visually how the viruses differ in Figure 4. The match between computations and length of genome is nice, but it is not clear what is the underlying basis for the agreement. Is the size of the base-paired length packed merely reflecting the accessible volume inside the capsid? The model is simple enough that one should be able to summarize what are the principal differences between the different viruses shown in Figure 4 (size of full capsid, charge of the arms, etc.).

2) Much is made of the observation that the viruses are overcharged, that isi.e., the ratio between the negative (nucleic acid) over the positive (protein arms) charges is larger than 1. Essentially, this means that the assembled capsid with its packed nucleic acid chain is overall negative. The text is emphatic: “our simulations show that ‘overcharged’ capsids are favored thermodynamically and kinetically, even in the absence of cellular factors or other external effects.” But while one does not dispute the outcome of the simulations, it is disappointing that no simple argument is provided nor sketched to explain *why* this is so (a back-of-the-envelope type of explanation). This would add much clarification on the underlying reasons for this outcome of the model. More importantly, it isn't clear that the charge ratio predicted by the model is correct or even reasonable. For instance, what about the likely presence of magnesium ions, which would presumably cancel out the charge of the excess of negative charges? The ionic screening model used here is Debye-Huckel and the model does not incorporate the possible effect of divalent binding. Furthermore, is it experimentally known that these capsids are negatively charged? This ought to be detectable through electrophoretic mobility measurements (zeta potential). A figure supplement provides a comparison of a calculation with explicit ions presented as a validation of the Debye-Huckel approximation (Figure 3—figure supplement 3). Not much detail is offered about this test; presumably this is for a 1:1 electrolyte in a primitive model representation of ionic solution (continuum dielectric solvent and hard-sphere ions), but no ion radii are given so it is hard to see how realistic this model is. And again, what about these divalent ions binding to the DNA?

3) Additional cause for concern about the treatment of ionic screening arise from Figure 2—figure supplement 2, which shows the residual excess chemical potential for adding one segment to the nucleic acid chain when it is in bulk solution (red line) or when it is in a capsid (blue line). What is noteworthy here is that the residual excess chemical potential is positive for both cases. Normally, insertion of one charged particle in a continuum solvent with an electrolyte is a negative number because the interaction of the charged particle with the surroundings, which comprises solvent and mobile counterions, is favorable. This excess free energy to add one chain segment in solution would probably be negative, even within the context of Debye-Huckel. We believe that the numbers in Figure 2—figure supplement 2 are positive because this interaction between the segment of the salt solution is not included in the present model. The latter probably only treats the Debye-Huckel screening of the interactions of the charged segment with other segments (which is positive and unfavorable) but the self-energy is neglected. Please confirm.

If this is so, the problem is that the favorable self-energy is probably much reduced once the segment is inserted inside the capsid because there is much less room to have counterions in the densely packed capsid. To take an analogy, this is a bit like an implicit solvent that changes the bare Coulomb's law q1*q1/r12 to q1*q2/(eps*r12), but which ignores the Born self-energy of the charges. Ignoring the self-term amounts to assume that it cancels out. This is okay if the environment is assumed to remain roughly the same. Is there enough free space left inside the capsid for the implicit solvent to account for the same Debye-Huckel treatment as in the bulk solution? What is the total volume of the interior of the capsid? What is the total fraction of volume occupied by the nucleic acid? How much free space is left for ionic solution (needed to justify the Debye-Huckel screening treatment)? Please clarify this issue and explain why the implicit treatment of counterions is sufficiently accurate here.

---

## [Author Response]

*1) Part of the appeal of the study is the simplicity of the underlying model, which confers great clarity to the analysis and the description of the results. The nicest and most impressive results are probably summarized in Figure 4. However, upon more scrutiny, one does also start to worry about the significance of the approximations that are being made, and how much confidence one has in the robustness of the conclusions. For example, it would be nice to convey visually how the viruses differ in Figure 4*.

We have modified the text (under “Predictions for specific viral capsid structures”) to direct the reader to Figure 1 where visualizations of the models are presented. We have only included images of one T=1 capsid and one T=3 capsid because the complex interior of the capsids makes it difficult to visually resolve factors other than capsid volume.

*The match between computations and length of genome is nice, but it is not clear what is the underlying basis for the agreement. Is the size of the base-paired length packed merely reflecting the accessible volume inside the capsid*?

As described later in the answer to point 3, the volume of the capsid is only one of several factors; we find that there is still a significant amount of free volume within the capsid when at the optimal packaging length. In order to further clarify how the structural features of the capsid lead to observed packaging, we have presented additional data as Figure 4—figure supplement 2. Please see the response to point 3 below for additional details.

*The model is simple enough that one should be able to summarize what are the principal differences between the different viruses shown in Figure 4 (size of full capsid, charge of the arms, etc.)*.

The differences between the capsid structures were summarized in a supplementary file. To better draw attention to this table, we have moved it to the main text (now Table 1). We have also added additional information (“Genome Length, Model Optimal Length, and Occupied Volume Fraction”).

*2) Much is made of the observation that the viruses are overcharged, i.e., the ratio between the negative (nucleic acid) over the positive (protein arms) charges is larger than 1. Essentially, this means that the assembled capsid with its packed nucleic acid chain is overall negative. The text is emphatic: “our simulations show that ‘overcharged’ capsids are favored thermodynamically and kinetically, even in the absence of cellular factors or other external effects.” But while one does not dispute the outcome of the simulations, it is disappointing that no simple argument is provided nor sketched to explain why this is so (a back-of-the-envelope type of explanation). This would add much clarification on the underlying reasons for this outcome of the model*.

The simulations show that overcharging is a consequence of two factors – the electrostatic screening and geometry. Namely, it is geometrically not possible for every charge on the encapsulated NA to approach within a Debye length of an opposite charge on the capsid. It is primarily the presence of these charges not able to interact with capsid charges that lead to overcharging. Earlier models were not able to capture this because the locations of charges were smeared by the assumption of spherical symmetry. Based on the reviewers’ question, we have modified the beginning of the Discussion to more clearly explain the origins of overcharging. We also present a new figure (Figure 5) to clarify this observation: if one accounts only for the NA charges that directly interact with capsid charges the system appears slightly undercharged (as predicted by the earlier continuum models with an assumption of spherical symmetry). The overcharging arises due to the presence of NA, which forms a path from capsid charge to capsid charge.

*More importantly, it isn’t clear that the charge ratio predicted by the model is correct or even reasonable. For instance, what about the likely presence of magnesium ions, which would presumably cancel out the charge of the excess of negative charges? The ionic screening model used here is Debye-Huckel and the model does not incorporate the possible effect of divalent binding. […] And again, what about these divalent ions binding to the DNA*?

Figure 3 and Figure 3—figure supplement 2 have been modified to include multiple sets of additional explicit ion simulations. The first set contains divalent cations as part of the salt solution. The second set of simulations includes divalent cations that are irreversibly bound to the NA (to represent specifically bound Mg^2+^ ions). These simulations indicate only a small effect due to divalent cations at physiological concentrations.

*Furthermore, is it experimentally known that these capsids are negatively charged? This ought to be detectable through electrophoretic mobility measurements (zeta potential)*.

The genome lengths of viruses assembled in vivo are known to high precision, as are the structures for the capsids we have considered. Based on these results it is incontrovertible that the genome length exceeds the positive capsid ARM charge. Given the complexity of the cellular environment, it could be argued that other multivalent cationic biomolecules might be present in these capsids. However, *in vitro* experiments mentioned in the text (e.g., Self-assembly of viral capsid protein and RNA molecules of different sizes: Requirement for a specific high protein/RNA mass ratio, J. Virol., 86(6):3318–3326, 2012; RNA encapsidation by SV40-derived nanoparticles follows a rapid two-state mechanism. J. Am. Chem. Soc., 134(21):8823–8830, 2012) have clearly shown that capsids spontaneously assemble into overcharged states under conditions in which the ionic composition is carefully controlled. Most notably, Cadena-Nava et al (J. Virol., 86(6):3318–3326, 2012) performed competition experiments, which showed that CCMV capsid proteins preferentially assemble around genomic length RNA rather than shorter fragments.

Zeta potential measurements do in fact show that capsids are negatively charged. However, these measurements do not evaluate overcharging of the capsid interior; rather, they are sensitive to the charge on the capsid exterior, which is separated by at least 5 nm from the interior. The capsid exterior is negatively charged because of acidic residues found there. (Since these negative charges on the capsid exterior are separated by at least 5 nm from the capsid interior, they have negligible effect on the thermodynamics of genome encapsulation. Therefore, we did not include them in our model.)

*A figure supplement provides a comparison of a calculation with explicit ions presented as a validation of the Debye-Huckel approximation (Figure 3—figure supplement 3). Not much detail is offered about this test; presumably this is for a 1:1 electrolyte in a primitive model representation of ionic solution (continuum dielectric solvent and hard-sphere ions), but no ion radii are given so it is hard to see how realistic this model is*.

We now completely describe the explicit ion (primitive model) simulations in the Methods section “Electrostatic interactions among polymer, ARM, and ion beads”. Furthermore, to fully understand the effect of excluded volume on our results, we have performed additional explicit ion simulations at various ion radii. The predicted dependence of optimal length on the ion radius is now shown in Figure 3—figure supplement 2. These simulations show that the optimal length predicted by the Debye-Huckel model is within 10% of that predicted by the primitive model with an ion radius of 0.125 nm (roughly the radius for Na^+^ and Cl^-^ ions in the CHARMM force field) at physiological salt concentration. We appreciate the reviewers for inspiring us to consider this further, and we have re-performed all the necessary explicit ion simulations using this realistic ion size, rather than the larger size (0.25 nm) we had used originally.

*3) Additional cause for concern about the treatment of ionic screening arise from Figure 2—figure supplement 2, which shows the residual excess chemical potential for adding one segment to the nucleic acid chain when it is in bulk solution (red line) or when it is in a capsid (blue line). What is noteworthy here is that the residual excess chemical potential is positive for both cases. Normally, insertion of one charged particle in a continuum solvent with an electrolyte is a negative number because the interaction of the charged particle with the surroundings, which comprises solvent and mobile counterions, is favorable. This excess free energy to add one chain segment in solution would probably be negative, even within the context of Debye-Huckel. We believe that the numbers in Figure 2—figure supplement 2 are positive because this interaction between the segment of the salt solution is not included in the present model. The latter probably only treats the Debye-Huckel screening of the interactions of the charged segment with other segments (which is positive and unfavorable) but the self-energy is neglected. Please confirm*.

*If this is so, the problem is that the favorable self-energy is probably much reduced once the segment is inserted inside the capsid because there is much less room to have counterions in the densely packed capsid. To take an analogy, this is a bit like an implicit solvent that changes the bare Coulomb’s law q1*q1/r12 to q1*q2/(eps*r12), but which ignores the Born self-energy of the charges. Ignoring the self-term amounts to assume that it cancels out. This is okay if the environment is assumed to remain roughly the same. Is there enough free space left inside the capsid for the implicit solvent to account for the same Debye-Huckel treatment as in the bulk solution? What is the total volume of the interior of the capsid? What is the total fraction of volume occupied by the nucleic acid? How much free space is left for ionic solution (needed to justify the Debye-Huckel screening treatment)? Please clarify this issue and explain why the implicit treatment of counterions is sufficiently accurate here*.

The excess chemical potential calculations were indeed performed using the Debye-Huckel model. We chose this model for these calculations because we had already found that simulations with the Debye-Huckel agree reasonably with those using the primitive model (e.g., Figure 3). The purpose of the chemical potential computations was to confirm the results of the calculations in which optimal length was estimated by rendering part of the capsid permeable to the encapsulated polymer. (While the thermodynamic justification for these latter calculations is straightforward, it is a somewhat novel approach to calculating the optimal length and thus worth testing by independent calculations.) We now clearly state that the excess chemical potential simulations were performed using the Debye-Huckel model in the Methods section “Equilibrium encapsidation”.

There is a significant amount of free space for counterions within capsids (both within our model and in actual capsids with single-stranded genomes). The most densely packed systems that we simulated are the T=1 biological capsids, which contain relatively long ARMs within a small capsid volume. For these models the fraction of *free volume* within the capsid is between 0.7–0.8. For the T=3 biological capsid models and for the simple capsid with ARM charge = +5, the free volume fraction is >0.9. These numbers indeed explain why the Debye- Huckel treatment is a reasonable starting point for modeling this system. We were remiss in not providing this justification for our model in the original manuscript; we now provide it in Table 1 and in the text of the Methods section.